# A Survey on Physiological Signal-Based Emotion Recognition

**DOI:** 10.3390/bioengineering9110688

**Published:** 2022-11-14

**Authors:** Zeeshan Ahmad, Naimul Khan

**Affiliations:** Department of Electrical, Computer and Biomedical Engineering, Toronto Metropolitan University, Toronto, ON M5B 2K3, Canada

**Keywords:** data annotation, physiological signals, data variability, emotion models, challenges, review

## Abstract

Physiological signals are the most reliable form of signals for emotion recognition, as they cannot be controlled deliberately by the subject. Existing review papers on emotion recognition based on physiological signals surveyed only the regular steps involved in the workflow of emotion recognition such as pre-processing, feature extraction, and classification. While these are important steps, such steps are required for any signal processing application. Emotion recognition poses its own set of challenges that are very important to address for a robust system. Thus, to bridge the gap in the existing literature, in this paper, we review the effect of inter-subject data variance on emotion recognition, important data annotation techniques for emotion recognition and their comparison, data pre-processing techniques for each physiological signal, data splitting techniques for improving the generalization of emotion recognition models and different multimodal fusion techniques and their comparison. Finally, we discuss key challenges and future directions in this field.

## 1. Introduction

Emotion is a psychological response to some external stimulus and internal cognitive processes, supported by a series of physiological activities going on in human body. Thus, emotion recognition is a promising and challenging work area which enables us to recognize the emotions of a person for stress detection and management, risk prevention, mental health and interpersonal relations.

Emotional response also depends on the age. Socio-cognitive approaches suggest that the ability to understand emotions should be well maintained in adult aging. However, neuropsychological evidence suggests potential impairments in processing emotions in older adults [1]. However, long term depression in any age can lead to chronic diseases [2]. The pandemic of COVID-19 affected the emotions of people across the globe. The prevalence of a high suicide risk increased from pre-pandemic to during the pandemic, appearing to be largely influenced by social determinants, in conjunction with the implications of the COVID-19 pandemic [3].

Emotion recognition is an emerging research area due to its numerous applications in our daily life. Applications include areas such as developing models for inspecting driver emotions [4], health care [5,6], software engineering [7] and entertainment [8].

Different modalities of data can be used for emotion recognition. They are commonly divided into behavioral and physiological modalities. Behavioral modalities includes emotion recognition from facial expressions [9,10,11,12,13], from gestures [14,15,16] and from speech [17,18,19], while physiological modalities include emotion recognition from physiological signals such as electroencephalogram (EEG), electrocardiogram (ECG), galvanic skin response (GSR), electrodermal activity (EDA) and so on [20,21,22,23,24].

Behavioral modalities can be effectively controlled by user and the quality of expressing them may be significantly influenced by personality of the user and the current environment of the subject [25]. On the other hand, physiological signals are continuously available and cannot be controlled intentionally or consciously. The commonly used physiological signals are ECG, EEG and GSR signals.

Many papers exist in literature where review on emotion recognition using physiological signals is presented.

In [26], a comprehensive review on physiological signal-based emotion recognition was presented that includes emotion models, emotion elicitation methods, the published emotional physiological datasets, features, classifiers and the frameworks for emotion recognition based on the physiological signal. In [27], a literature review provides a concise analysis of physiological signals and instruments used to monitor them, emotion recognition, emotion models and emotional stimulation approaches. The authors also discuss selected works on emotional recognition by physiological signals in wearable devices. In [28], different emotion recognition methods using physiological signals by machine learning techniques were explained. This paper also reviewed different stages such as data collection, data processing, feature extraction and classification models for each recognition method. In [29], recent advancements in emotion recognition research using physiological signals, including emotion models and stimulation, pre-processing, feature extraction and classification methodologies were presented. In [30], the current state-of-the art of emotion recognition was presented. The paper also presented the main challenges and future opportunities that lie ahead, in particular for the development of novel machine learning (ML) algorithms in the context of emotion recognition using physiological signals.

In [31], the different stages of Facial Emotion Recognition (FER) such as pre-processing, feature extraction and classification using various methods and state-of-the-art CNN models are discussed. Comparison between different deep learning models, their benchmark accuracy and their architectural details are also discussed for model selection based on application and dataset. In [32], the authors present a systematic literature review of scientific studies investigating automatic emotion recognition in a clinical population composed of at least a sample of people with a disease diagnosis. Based on the findings it is revealed that most clinical applications involved neuro-developmental, neurological and psychiatric disorders with the aims of diagnosing, monitoring, or treating emotional symptoms. In [33], emotion recognition for everyday life using physiological signals from wearables is presented. The authors observed that deep learning architectures provide new opportunities to solve complex tasks in this field of study. However, the limitation is that the study presents classification of binary or few class problem. In [34], the current situation in the EEG-based emotion recognition research along with the tutorial is presented to guide the researchers to start from a very beginning, as well as illustrate the theoretical basis and the research motivation. EEG data pre-processing, feature engineering, selection of classical and deep learning models for EEG-based emotion recognition are discussed.

In [35,36,37], the neural network, deep learning models and transfer learning-based models were discussed for different physiological signals and physiological signal-based datasets.

Existing review papers on emotion recognition based on physiological signals looked for only the regular steps involved in the workflow of emotion recognition such as pre-processing, feature extraction and classification but did not discuss the set of important challenges that are particular to emotion recognition.

The existing review papers on emotion recognition based on physiological signals provide only the following information:feature extraction and selection techniques;generic data pre-processing techniques for physiological signals;different types of classifiers and machine learning techniques used for emotion recognition;databases for emotion recognition;assessment and performance evaluation parameters for ML models such as calculation of accuracy, recall, precision and $F_{1}$ score from confusion matrix.

Although the above information is useful and important for emotion recognition, it is established for any signal processing application and thus missing those challenging factors that are specific for emotion recognition using physiological signals. The following challenging factors are not discussed in the existing literature.

The problems faced during data annotation of physiological signals are not elaborated.The data pre-processing techniques are bundled together and presented as generic techniques for any physiological signal. In our opinion, each physiological signal presents its own unique set of challenges when it comes to pre-processing, and the pre-processing steps should be discussed separately.Inter-subject data variability has a huge impact on emotion recognition. The existing reviews neither discuss this effect nor provide recommendations to reduce inter-subject variability.A comparison of data splitting techniques, such as subject-independent and subject-dependent, is not provided for better emotion recognition and generalization of the trained classification models.The comparison and advantages of different multimodal fusion methods are not provided in these review papers.

The aforementioned challenging factors for emotion recognition that are missed in the existing literature are considered as the research gaps of the existing work. Thus, to bridge the above gap and to address the shortcomings of the existing reviews, in this paper, we are focussed on the missing but essential elements of emotion recognition. These are:different kinds of data annotation methods for emotion recognition and advantages/disadvantages of each method;data pre-processing techniques that are specific for each physiological signal;effect of inter-subject data variance on emotion recognition;data splitting techniques for improving the generalization of emotion recognition models;different multimodal fusion techniques and their comparison.

This paper is organised as follows: in Section 2, the emotion models are explained. In Section 3, databases for emotion recognition using physiological signal are presented. In Section 4, the data annotation techniques for emotion recognition are illustrated. Section 5 demonstrates the data pre-processing technique for each physiological signal. In Section 6, the effect of inter-subject data variance on emotion recognition is described. In Section 7, data splitting techniques for improved generalization of emotion recognition models are analyzed. Section 8 provides the details of different multimodal fusion techniques for emotion recognition. Section 9 summarizes future challenges of emotion recognition using physiological signals and finally we conclude the paper in Section 10.

The organisation of the paper is also shown in Figure 1.

## 2. Emotion Models

Over the past few decades, different emotion models have been proposed by researchers based on the quantitative analysis and the emergence of new emotion categories. Based on recent research, emotional states can be represented with two models: discrete and multidimensional.

### 2.1. Discrete or Categorical Models

Discrete models are the most commonly used models because they contain a list of distinct emotion classes that are easy to recognize and understand. Ekman [38] and Plutchik [39] are amongst those scientists that present the concept of discrete emotion states. Ekman briefed that there are six basic emotions—happy, sad, anger, fear, surprise, and disgust—and all the other emotions are derived from these six basic emotions. Plutchik presented a famous wheel model to describe eight discrete emotions. These emotions are joy, trust, fear, surprise, sadness, disgust, anger and anticipation, as shown in Figure 2. The model describes the relations between emotion concepts, which are analogous to the colors on a color wheel. The cone’s vertical dimension represents intensity, and the circle represents degrees of similarity among the emotions. The eight sectors are designed to indicate that there are eight primary emotion dimensions and rest of the motion dimensions are derived from them.

### 2.2. Continuous or Multidimensional Models

All emotional states listed in discrete sets of emotions cannot be confined by a single word and they need a range of intensities for description. For instance, a person may feel less or more excited, or become less or more afraid in response to a particular stimulus. Thus, to cover the range of intensities in emotions, multidimensional models such as 2D and 3D are proposed. Amongst 2D models, the model presented in [40] is the famous model that describes the emotion along the dimensions of High Arousal (HA) and Low Arousal (LA) and High Valence (HV) and Low Valence (LV) as shown in Figure 3. The 2D model classifies emotions based on two dimensional data consisting of valence and arousal value. On the other hand, the 3D model deals with valence, arousal and dominance. Valence indicates the level of pleasure, Arousal indicates the level of excitation and Dominance indicates the level of controlling or dominating emotion. The 3D model is shown in Figure 4.

## 3. Databases

Several datasets for physiological-based emotion recognition are publicly available, such as AMIGOS [41], ASCERTAIN [42], BIO VID EMO DB [43], DEAP [44], DREAMER [45], MAHNOB-HCI [46], MPED [47], SEED [48] as shown in Table 1. All datasets, other than SEED, are multimodal and provide more than one modality of physiological signals. As can be seen, some of these datasets utilize the continuous emotion model, while the others use the discrete model. Most of these datasets are unfortunately limited to a small number of subjects due to the elaborate process of data collection. Some details of these datasets such as annotation and pre-processing are discussed in the following sections.

## 4. Data Annotation

Data annotation is amongst the most important steps after data acquisition and eventually for emotion recognition. However, this subject is not addressed in detail in existing literature. For better discussion on this matter, we divide the data annotation procedure into discrete and continuous data annotation.

In discrete annotation, participants are given questionnaire after the stimuli and they are asked to rate their feelings between some scores ranging from 1 to 5 or from 0 to 9 and then the final annotations are decided as shown in Figure 5.

On the other hand, in continuous annotations, the participants are required to continuously annotate the data in real time using some human computer interface (HCI) mechanism as shown in Figure 6.

In Section 4.1 and Section 4.2, we provide a review on discrete and continuous annotation.

### 4.1. Discrete Annotation

In [47], three psychological questionnaires—PANAS [50], SAM [51] and DES [52]—were used for data annotation of seven emotion categories such as joy, funny, anger, sad, disgust, fear and neutrality. Besides questionnaires, a T-test was also conducted to evaluate the effectiveness of the selected emotion categories.

In [42,53], participants were briefed on details of the experiments including the principal experimental tasks and later on the self-report questionnaires were filled by the participants according to their experiences during stimuli. These questionnaires were finally used for data annotation. In [22], for automatic ECG-based emotion recognition, music is used as stimuli and in the self-assessment stages, the participants were asked to indicate how strongly they had experienced the corresponding feeling during stimulus presentation and then participants gave marks ranging from 1 to 5 for each of the nine emotion categories.

In [54], the questionnaire, divided into three parts, one for each activity, in each of which the participants had to select the emotions they felt before, during and after the activity, was given to the participants. Based on the participants marking, 28 emotions are arranged on a 2-dimensional plane with Valence and Arousal at each axis as shown in Figure 3, according to the model proposed in [55]. In [56], the data were collected in a VR environment. The data annotation was based on the subjects’ responses to the two questions regarding level of excitement and pleasantness which were graphed on a 9 square grid, a discretized version of the arousal-valence model [57]. Each grid shows how many subjects reported feeling the corresponding level of “pleasantness” and “excitement” for each VR session. The valence scale was measured as “Unpleasant”, “Neutral”, or “Pleasant” (from left to right) and the arousal scale was measured as “Relaxing”, “Neutral”, or “Exiting” (from bottom to top). The results are very asymmetrical with the majority of the sessions rated as “Exciting” and “Pleasant”.

### 4.2. Continuous Annotation

In [49], participants viewed a 360° videos as stimuli and they rated emotional states (valence and arousal) continuously using the joystick. It is observed that collecting continuous annotations can be used to evaluate the performance of fine-grained emotion recognition algorithms such as weakly supervised learning or regression. In [58], the authors claim that the lack of continuous annotations is the reason why they failed to validate their weakly-supervised algorithm for fine-grained emotion recognition since the continuous labels allow the algorithms to learn the precise mappings between the dynamic emotional changes and input signals. Moreover, if only discrete annotations are available, ML algorithms can overfit because the discrete labels represent only the most salient or recent emotion rather than the dynamic emotional changes that may occur within stimuli [59]. A software for continuous emotion annotation is introduced in [60] and is called EMuJoy (Emotion measurement with Music by using a Joystick). It is better than Schubert’s 2DES software [61]. It helps subjects to generate self reports for different media in real time with a computer mouse, joystick, or any other human–computer interface. Using a joystick has an advantage, because joysticks have a return spring to automatically realign them in the middle of the space. This contrasts with Schubert’s software, where mouse is used. In [62], stimuli videos were also continuously annotated along valence and arousal dimensions. Long-short-term-memory recurrent neural networks (LSTM-RNN) and continuous conditional random fields (CCRF) were utilized in detecting emotions automatically and continuously. Furthermore, the effect of lag on continuous emotion detection is also studied. The database is annotated using a joystick and delays from 250 ms up to 4 s in annotations. It is observed that 250 ms increased the detection performance whereas longer delays deteriorated it. This shows the advantage of using joystick over mouse. The authors of [63] analyzed the effect of lag on continuous emotion detection on SEMAINE database [64]. They found a delay of 2 s will improve their emotion detection results. SEMAINE database is annotated by Feeltrace [65] using a mouse as annotation interface. Thus, the observations in [63] also show that the response time of joystick is less than mouse. Another software named DARMA is introduced in [66] for continuous measurement system that synchronizes media playback and the continuous recording of two-dimensional measurements. These measurements can be observational or self-reported and are provided in real-time through the manipulation of a computer joystick.

Touch events are dynamic and can be bidirectional, i.e., 2D along valence-arousal plane. During continuous annotation, participants are instructed to annotate their emotion experience by moving the joystick head into one of the four quadrants. The movement of the joystick head maps the emotions into a 2D valence-arousal plane, in which the x axis indicates valence while the y axis indicates arousal. Since the nature of emotions is time-varying, during annotation, participants could lose the control over the speed, i.e., the movement of a joystick which could lead collection of less precise ground truth labels. Thus, the training of participants on HCI mechanism and study of lag for continuous emotion detection are two important factors for precise continuous annotation.

### 4.3. Summary of the Section

There is no definite or clear-cut advantage of discrete annotation on continuous annotation and vice versa. However, the choice of annotation method depends on the application. For example, in biofeedback systems, participants learn how to control physical and psychological effects of stress using feed back signals. Biofeedback is a mind–body technique that involves using visual or auditory feedback to teach people to recognize the physical signs and symptoms of stress and anxiety, such as increased heart rate, body temperature and muscle tension. Since stress level changes during biofeedback mechanism, real time continuous annotation is more important in this case [67]. The choice of annotation also depends on length of the data, the number of emotion categories, number of participants, feature extraction method and the choice of classifier.

## 5. Data Pre-Processing

Physiological signals such as electroencephalogram (EEG), electrocardiogram (ECG) and galvanic skin response (GSR) are time-series and delicate signals in the raw form. During acquisition, these physiological signals are contaminated by numerous factors such as line interference of 50 Hz or 60 Hz, electromagnetic interference, noise and baseline drifts and different artifacts due to body movements and different responses of participants to different stimuli. The bandwidths of the physiological signals are different. For instance, EEG signals lie in the range of 0.5 Hz to 35 Hz, ECG signals are intensive in the range of 0 to 40 Hz and GSR signals are mainly concentrated in the range of 0 to 2 Hz. Since the strength of physiological signals is rich in different frequency ranges, different physiological signals required different pre-processing techniques. In this section, we review the pre-processing techniques used for physiological signals.

### 5.1. EEG Pre-Processing

In addition to noise, the EEG signals are affected by Electrooculography (EOG) with frequency below 4 Hz. This artifact is caused by the facial muscle movement or eye movement [68,69]. Different methods that have been used for EEG pre-processing or cleaning include the rejection method, linear filtering, statistical methods such as linear regression and Independent Component Analysis (ICA).

#### 5.1.1. Rejection Method

The rejection method involves both manual and automatic rejection of epochs contaminated with artifacts. Manual rejection needs less computations but it is more laborious than automatic rejection. In automatic rejection pre-determined threshold is used to remove the artifact-contaminated trials from the data. The common disadvantages associated with the rejection method are the sampling bias [70] and loss of valuable information [71].

#### 5.1.2. Linear Filtering

Linear filtering is simple and easy to apply and is beneficial mostly when artifacts located in certain frequency bands do not overlap with the signal of interest. For example, low-pass filtering can be used to remove EMG artifacts and high-pass filtering can be used to remove EOG artifacts [48]. However, linear filtering flops when EEG signal and the EMG or EOG artifacts lie in the same frequency band.

#### 5.1.3. Linear Regression

Linear regression using least square method has been used for EEG signal processing for removing EOG-based artifacts. In linear regression, using least square criteria, residual is calculated by subtracting the EOG signal from the EEG signal and then this residual is optimized to get the cleaned EEG signal [72]. This method is not suitable for EMG-based artifact removal because EMG data are collected from multiple muscles groups and therefore EMG data has no single reference.

#### 5.1.4. Independent Component Analysis

ICA is amongst those methods that do not require reference artifacts for cleaning the EEG signal and thus making the components independent [73]. The disadvantage associated with ICA is that it requires prior visual inspection to identify the artifact part.

### 5.2. ECG Pre-Processing

ECG signal is usually corrupted by various noise sources and artifacts. These noises and artifacts deteriorate the signal quality and affect the detection of QRS.

#### 5.2.1. Filtering

Filtering is the simplest method of removing noise and artifacts from the ECG signal and to improve the signal to noise ratio (SNR) of the QRS complex. In ECG signal, the three critical frequency regions include the very low frequency band below 0.04 Hz, the low frequency band (0.04 to 0.15 Hz), and the high frequency band (0.15 to 0.5 Hz) [74]. Thus the commonly used filters for ECG are high pass filter, low pass filter and band pass filter.

High-pass filters allow only higher frequencies to pass through them. They are used to suppress low-frequency components the ECG signal. Low frequency components include motion artifact, respiratory variations and baseline wander. In [75], high pass filters with cut-off frequency of 0.5 Hz was selected to waive off the baseline wander.

Low-pass filters are usually employed to eliminate high-frequency components from the ECG signal. These components include noise, muscle artifacts and powerline interference [76].

Since bandpass filters remove most of the unwanted components from the ECG signal, it is extensively used to pre-process ECG. The structure of Band-pass filter comprises of both a high-pass and low-pass filter. In [77], bandpass filter is used to remove muscle noise and baseline wander from ECG.

#### 5.2.2. Normalization

In ECG, the amplitude of QRS complex amplitude increase from birth to adolescence and then begins to decrease afterward [78]. Due to this, ECG signals suffer from inter-subject variability. To overcome this problem, amplitude normalization is performed. In [79], a method of normalizing ECG signals to a standard heart rate to lower the false rate detection, was introduced. The commonly used normalization techniques are min-max normalization [80], maximum division normalization [81] and Z-score normalization [82].

### 5.3. GSR Pre-Processing

There is still no standard methods for GSR pre-processing. In this section, we are discussing a few methods to deal with noise and artifacts present in raw GSR signal.

#### 5.3.1. Empirical Mode Decomposition

Empirical Mode Decomposition (EMD) is the technique that best addresses the nonlinear and nonstationary nature of the signal while removing noise and artifact and was introduced in [83] as a tool to adaptively decompose a signal into a collection of AM–FM components. The EMD relies on a fully data-driven mechanism that does not require any a priori known basis, like Fourier and wavelet-based mechanisms. EMD decomposes the signal into a sum of intrinsic mode functions (IMFs). An IMF is defined as a function with equal number of extrema and zero crossings (or at most differed by one) with its envelopes, as defined by all the local maxima and minima, being symmetric with respect to zero.

After extraction of IMFs from a time series signal the residue tends to become a monotonic function, such that no more IMFs can be extracted. Finally, after the iterative process, the input signal is decomposed into a sum of IMF functions and a residue.

In [84], an algorithm, modified from EMD, called Empirical iterative algorithm is proposed for denoising the GSR from motion artifacts and quantization noise as shown in Figure 7 [84]. The algorithm does not rely on the shifting process of EMD, and provides the filtered signal directly as an output of each iteration.

#### 5.3.2. Kalman Filtering

Low pass and moving average filters have been used for GSR pre-processing [29]; however, the Kalman filter is the better choice. Kalman filter is a model-based filter and a state estimator. It employs a mathematical model of the process producing the signal to be filtered [85]. In [86], an extended Kalman filter is used to remove noise and artifact from GSR signal in real time. The Kalman filter used in this article is comparable to a third order Butterworth low pass filter with a similar frequency response. However, the Kalman filter is more robust than its counterpart Butterworth low pass filter while suppressing the noise and artifacts.

### 5.4. 1D to 2D Conversion

Physiological signals such as EEG, ECG and GSR are 1D signal in raw form. In [87,88], raw ECG signal was converted into 2D form, i.e., into three statistical images namely Gramian Angular Field (GAF) images, Recurrence Plot (RP) images and Markov Transition Field (MTF) images as shown in Figure 8. Experimental results show the superiority of 1D to 2D pre-processing. In [89], ECG and GSR are converted in 2D scalogram for better emotion recognition. Furthermore, in [90], ECG and GSR signals are converted in 2D RP images for improved emotion recognition as compared to 1D form signals.

Thus, the transformation from 1D to 2D has been shown to be an important pre-processing step for physiological signals.

### 5.5. Summary of the Section

The pre-processing for the EEG data is mainly to remove EOG artifacts with a frequency less than 4 Hz that caused by eye blink, EMG artifacts with a frequency more than 30 Hz, power frequency artifacts in the environment with a frequency between 50 to 60 Hz and so on. Since the noise and artifacts present at different frequency ranges, therefore linear filtering, linear regression and ICA are more useful pre-processing methods for EEG. The other methods such as EMD decomposes the signal into a sum of intrinsic mode functions (IMFs). This may supress the information from EEG data. Similarly, the limitations of Kalman filtering are that it is like a third order Butterworth low pass filter and thus cannot remove noise from all the frequency bands of EEG.

## 6. Inter-Subject Data Variance

In this section, we will review the effect of inter-subject data variance on emotion recognition. We will also discuss the reasons of inter-subject data variance and recommend solutions for the problem.

There exists few papers [42,53,91,92,93,94], which only reported the problem of inter-subject data variance and its effect on degradation of emotion recognition accuracy, but they did not shed light on the possible reasons behind inter-subject data variability.

To derive our point home, we performed experiments on ECG data of two datasets, WESAD dataset [95] and our own Ryerson Multimedia Research Laboratory (RML) dataset [96]. For WESAD dataset, we use three categories. These categories are Amusement, Baseline and Stress. For RML dataset, three categories of stress are low stress, medium stress and high stress. The issue of inter-subject data variability was particularly apparent in the RML dataset, since we controlled the data collection process.

To observe the high inter-subject variability, we utilize leave-one-subject-out cross-validation (LOSOCV) for all our experiments, where the models are trained with data from all but one subject, and tested on the held out subject. The average results across all subjects are presented in Table 2, Table 3, Table 4 and Table 5. Table 2 and Table 4 show the results of experiments conducted with 1D ECG for both datasets. To see the affect of data pre-processing on inter-subject data variability, we transform the raw ECG data into spectrograms. ResNet-18 [97] was trained on these spectrograms and results are shown in Table 3 and Table 5.

In Table 2, Table 3, Table 4 and Table 5, we observe the inter-subject data variance across all the metrics, i.e., accuracy, precision, recall and $F_{1}$ score. We also observe that by transforming 1D data to 2D using spectrograms have increased the accuracy of the affective state but did not reduce the data variance significantly.

Looking at the WESAD results, we can see that the average accuracy is only 72%. This is significantly lower than the average accuracy achieved if we would use k-fold validation. However, using k-fold cross validation means that a segment of data for each subject was always present in the training set. This is an unrealistic scenario for practical applications. Therefore, dealing with inter-subject variability, conducting experiments in a LOSOCV setting is very important. We elaborate on this point further in Section 7.

### 6.1. Statistical Analysis

We perform statistical analysis by plotting error-bars on 1D and 2D WESAD data to notice the variability in accuracy of each individual subject across the overall mean value of accuracy as shown in Figure 9 and Figure 10. The error-bars show the spread or variability in the data. Figure 9 and Figure 10 can also be used to detect the outlier subjects from the dataset that could be removed later on for reducing the variability amongst the remaining subjects.

### 6.2. Reasons of Inter-Subject Data Variance

The following reasons could explain the inter-subject variability:Some users feel uncomfortable when wearing a sensor on their bodies and behave unnaturally and unexpectedly during data collection.Body movements during data collection adversely effect the quality of recorded data. For instance, head and eyeball movement during EEG data collection and hand movement during EMG data collection are the sources of degradation of these data.An intra-variance subject is also caused due to the placement of electrodes on different areas of the body. For example, during chest-based ECG collection, electrodes are placed on different areas of the chest and during collection of EEG data, the electrodes are placed on all over the scalp, i.e., from left side to right side through the center of the head.Different users respond differently to the same stimuli. This difference in their response depends on their mental strength and physical stability. For instance, a movie clip can instill fear in some users while other users may feel excitement while watching it.The length of the collected data is also amongst the major reasons of creating subject diversity. A long video is more likely to elicit diverse emotional states [98], while the data with limited samples will create problem during the training of the model and classification of the affective states.Experimental environment and quality of sensors/equipment are also the reasons of variance in data. For instance, uncalibrated sensors may create errors at different stages of data collection.

### 6.3. Reducing Inter-Subject Data Variance

The following actions could be taken to remove the inter-subject variability:The collected sample size for each subject should be same. The length of the data should be set carefully to avoid variance in the data.The users or subjects should be trained to remain calm and behave normally during data collection or wearing a sensor.Different channels of the data collecting equipment should be calibrated against the same standards so that the intra-variance in the data for the subject can be reduced.In [99], statistics-based unsupervised domain adaptation algorithm was proposed for reducing the inter-subject variance. A new baseline model was introduced to tackle the classification task, and then two novel domain adaptation objective functions, cluster-aligning loss and cluster-separating loss were presented to mitigate the negative impacts caused by domain shifts.A state-of-the-art attention-based parallelizable deep learning model called transformer was introduced in [100] for language modelling. This model uses multihead attention to focus more on relevant features for improved language modelling and translational task. A similar kinds of attention-based deep learning models should be developed for physiological signals for reducing intersubject variability.

### 6.4. Summary of the Section

The following points are the key findings of the section:Inter-subject data variance depends on the behavior of the user during data collection, calibration of the sensors and experimental environment.By performing experiments, we prove that inter-subject data variance could be reduced by data pre-processing, i.e., by transforming the raw ECG data (1D) into spectrograms (2D). Transforming data from 1D to 2D enables ResNet-18 to extract those relevant features that are more specific to emotions.We believe more research in the area of domain adaptation and developing attention-based models could help tackle inter-subject data variability.

## 7. Data Splitting

There are two major types of data splitting techniques that are being used for emotion recognition using physiological signals. These two methods of splitting are called subject-independent and subject-dependent.

In the subject-independent case, the model is trained from data of various subjects and then tested on data of new subjects (that are not included in the training part). In the subject-dependent case, the training and testing part is randomly distributed among all the subjects.

### 7.1. Subject-Independent Splitting

The most commonly used subject-independent data splitting method is leave-one-subject-out (LOSO) cross-validation. In LOSO, the model is trained on (k −1) subjects and leave one subject for testing and this process is repeated for every subject of the dataset.

In [101,102], LOSO cross validation technique is used for DEAP dataset containing 32 subjects. Thus, the experiment is repeated for every subject and the final performance is considered as the average of all experiments. In [103], leave-one-subject out method is conducted to evaluate the performance, and the final performance is calculated by the averaging method. In [104], EEG-based emotion classification was conducted by a leave-one-subject out validation (LOSOV). One subject among the 20 subjects is used for testing, while the others were used for the training set. The experiments were repeated for all subjects and the mean and standard deviation of the accuracies were used to evaluate the model performance.

### 7.2. Subject-Dependent Splitting

In subject-dependent splitting, various strategies are used. In one of the strategies, training and testing parts are taken from a single subject with a particular splitting percentage. In another setting, few trials of same subject are taken as a training part and rest of them are used as testing. Furthermore, in another method of subject-dependent splitting, testing and training parts are randomly selected across all the subjects either by particular percentage or by k-fold cross validation. For k = 5, the concept of cross validation is shown in Figure 11. The final performance is the average of all the five evaluations.

In [47], physiological signals of 7 trials out of 28 were selected as testing for each subject and rest of the trials are used for training. This was repeated for every subject. In [105], models are evaluated using 10-fold cross validation performed on each subject. This was repeated on all 32 subjects of DEAP datasets and the final evaluation about accuracies and *F*_1_ scores are based on averaging method. In [106], physiological samples of all subjects of DEAP dataset are used to evaluate the model. The 10-fold cross validation technique is used to analyze the emotion recognition performance. All samples are randomly divided into ten subsets. One subset is regarded as a test set, and another subset as a validation set. The remaining subsets are regarded as a training set. The above process is repeated ten times until all subsets are tested.

### 7.3. Summary of the Section

Subject-dependent classification is usually performed to investigate the individual variability between subjects on emotion recognition from their physiological signals.

In real application scenarios, emotion recognition system with subject-independent classification is considered more practical as the system has to predict emotions of each subject or patient separately when implemented for emotion recognition task.

In [94], comparison between subject-dependent and subject-independent setting is provided. it is observed that subject-independent splitting shows 3% lower accuracy because of high inter-subject variability. However, the generalized capability of subject-independent model on unseen data is more as these models learn inter-subject variability.

Thus, it is more valuable to develop a good subject-independent emotional recognition model so that they can generalize well on unseen data or new patients.

## 8. Multimodal Fusion

The purpose of multimodal fusion is to obtain complementary information from different physiological modalities to improve the emotion recognition performance. Fusing different modalities alleviates the weaknesses of individual modalities by integrating complementary information from the modalities.

In [107], new emotional discriminative space is constructed utilizing discriminative canonical correlation analysis (DCCA) is used from physiological signals with the assistance of EEG signals. Finally, machine learning techniques are utilized to build emotion recognizer. The authors of [108] trained feedforward neural networks using both the fused and non-fused signals. Experiments show that the fused method outperforms each individual signal across all emotions tested. In [109], feature level fusion is performed between the features from two modalities, i.e., facial expression images and EEG signals. A feature map of size 128×128 is constructed by concatenating the facial images and their corresponding EEG feature maps. Emotion recognition model is finally trained on these feature maps. Experimental results show the improved performance of multimodal features over single modality features. In [110], hybrid fusion of face data, EEG and GSR is performed. First, features from EEG and GSR modalities are integrated for estimating the level of arousal. Final fusion was the late fusion of EEG, GSR and face modalities.

The authors of [111] presented a new emotion recognition framework based on decision fusion, where three separate classifiers were trained on ECG, EMG and skin conductance level (SCL). The majority voting principle was used to determine a final classification result on the three outputs of the separate classifiers.

In [112], multidomain features such as features from time domain, frequency domain and wavelet domain are fused with different combinations to identify stable features which would best represent the underlying EEG signal characteristics and performed better classification. Multidomain feature fusion is performed using concatenation to obtained final feature vector. In [113], weighted average fusion of physiological signal was conducted to classify valence and arousal. Overall performance was quantified as the percentage of correctly classified emotional states per video. It is also observed that the classification performance obtained was slightly better for arousal than valence. The authors of [114] provide the study about combining the EEG data and audio-visual features of the video. Then, PCA algorithm was applied to reduce the feature dimensions. In [115], feature level fusion of sensor data from EEG and EDA sensors was performed for human emotion detection. Nine features (delta, theta, low alpha, high alpha, low beta, high beta, low gamma, high gamma and galvanic skin response) are selected for training a neural network model. The classification performance of each modality was also evaluated separately. It is observed that multimodal emotion recognition is better than using a single modality. In [116], the authors developed two late multi-modal data fusion methods with deep CNN models to estimate positive and negative affects (PA and NA) scores and explored the effect of the two late fusion methods and different combination of the physiological signals on the performance. In the first late fusion method, a separate CNN is trained for each modality and in the second late fusion method, the average of the three classes’ (baseline, stress and amusement) probabilities across the pretrained CNNs is calculated, and the emotion class with the highest average probability is selected. In the case of the PA or NA estimation, the average of the estimated scores across the pretrained CNNs is provided as the estimated affect score. In [96], multidomain fusion of ECG signal is performed for multilevel stress assessment. First, the ECG signal was converted into image and then images are made multimodal and multidomain by transforming them into frequency and time-spectrum domain using Discrete Fourier Transform (DFT) and Gabor wavelet transform (GWT) respectively. Features in different domains are extracted by CNNs and then decision level fusion is performed for improving the classification accuracy as shown in Figure 12.

### Summary of the Section

The above review shows that the fusion of the physiological modalities exhibits better performance than the single modality in terms of classification accuracy for both arousal and valence dimensions. Furthermore, it is observed from the review that the two major fusion methods practiced for physiological data features are feature fusion and decision fusion; however, feature fusion is adopted by the researcher more than decision level. The greatest advantage of feature level fusion is that it utilizes the correlation among the modalities at an early stage. Furthermore, only one classifier is required to perform a task, making the training process less tedious. The disadvantage with decision level fusion is the use of more than one classifier. This makes the task time consuming.

## 9. Future Challenges

The workflow for emotion recognition consists of many steps such as data acquisition, data annotation, data pre-processing, feature extraction and selection and recognition. The problems and challenges are in each step of emotion recognition and are explained in this section.

### 9.1. Data Acquisition

One of the significant challenges in acquiring physiological signals is to deal with noise, baseline drifts, different artifacts due to body movements, different responses of participants to different stimuli and low graded signal. The data acquiring devices carry noise which corrupts the signal and induced artifacts that superimposed on the signal. The second challenge is the setting of stable and noiseless lab and selection of stimuli so that genuine emotions are induced which are closed to real world feelings. Another challenge in acquiring a high quality data is the various responses of participants or subjects to the same stimuli. This causes large inter-subject variability in the data and is also the reason that most of the available emotion recognition datasets are of short duration.

To overcome the above challenges, well-designed labs and proper selection of stimuli are necessary. Furthermore, subjects must be properly trained to avoid large variance in the data and possibility of gathering datasets with long duration and less inter-subject variability.

### 9.2. Data Annotation

High volume research has been conducted on emotion recognition, even then there is no uniform standards to annotate data. Discrete and continuous data annotation techniques are commonly used; however, due to unavailability of uniform annotation standard, there is no compatibility between the datasets. Furthermore, all emotions can neither be listed as discrete nor continuous as different emotions need different range of intensities for description. The problem with existing datasets is that some of them are annotated with four emotions, some are with basic six emotions or with eight emotions. Due to this different emotion labels, datasets are not compatible and emotion recognition algorithms do not work equally well on these datasets.

To face the aforementioned challenges of data annotation, hybrid data annotation technique should be introduced where each emotion is labelled according to its range of energy. However, care must be taken while selecting new standards of data annotation because hybrid annotation technique may lead to imbalance data.

### 9.3. Feature Extraction and Fusion

Features for emotion recognition are categorized into time domain features and transform domain features. Time domain features mostly include statistical features of first and higher orders. Transform domain features are further divided into frequency domain and time-frequency domain such as feature extracted from wavelet transform. Still, there is no set of features that guarantees to work for all models and situations.

One of the solution of the above problem is to design adaptive intelligent systems that can automatically select the best features for the classifier model. Furthermore, another solution is to perform multimodal fusion. In Section 8, it is explained that commonly practiced fusion techniques for emotion recognition are feature level and decision level fusion. However, for feature level fusion, concatenation is mostly used and for decision level fusion, majority voting is mostly practiced. Thus, a need for more improved and intelligent-based fusion methods is arising because the simple feature level and decision level fusion cannot counter the non-stationary and nonlinear nature of features.

### 9.4. Generalization of Models

One of the biggest challenges in emotion recognition is to design the models that can generalize well on unseen data or new datasets. The main obstacles are the limited samples in the datasets, non-standard data splitting techniques and large inter-subject data variability. The commonly used data splitting techniques are subject-dependent and subject-independent techniques. It is explained in Section 7 that the models trained on subject-independent settings are more likely to generalize well. However, existing subject-independent recognition models are not intelligent enough to perform well in realistic and real-time applications.

Since, while testing, an emotion recognition model has to face an unseen subject, one solution is to train and validate the model on large number of subjects so that it can generalize well on testing. The problem of a limited dataset can be solved by carefully applying data augmentation techniques because that techniques could lead to data imbalance and overfitting of the model.

### 9.5. Modern Machine Learning Methods

Advanced machine learning tools need to be developed to mitigate the challenges of emotion recognition using physiological signals. For instance, one of the modern techniques which is being used is transfer learning. In transfer learning, the knowledge learned while solving one task can be applied for different but related work. For example, the rich features learned by a machine learning model while conducting emotion recognition based on EEG data from a large dataset could also be applicable for EEG data of another dataset. This can easily solve the problem of a small dataset where fewer data samples are available. Transfer learning methods are getting success but still lot of research work is required to be conducted for physiological signal-based transfer learning methods for emotion recognition.

## 10. Conclusions

In this paper, we provide a review on the physiological signal-based emotion recognition. Existing reviews on the physiological signal-based emotion recognition presented only the generic steps of emotion recognition such as combined techniques for data pre-processing, feature extraction and selection methods, selection of machine learning techniques and classifiers, but did not elaborate on the most important factors that are crucial for the performance of emotion recognition systems and their generalization. These important factors include the challenges during data annotation, specific data pre-processing techniques for each physiological signal, effect of inter-subject data variance, data splitting methods and multimodal fusion. Thus, in this paper, we address these all challenging factors to bridge the gap in the existing literature. In this research, we provide comprehensive review on these factors and report our key findings about each factor. We also discuss the future challenges about physiological signal-based emotion recognition based on this research.

## Figures and Tables

**Figure 1 bioengineering-09-00688-f001:**
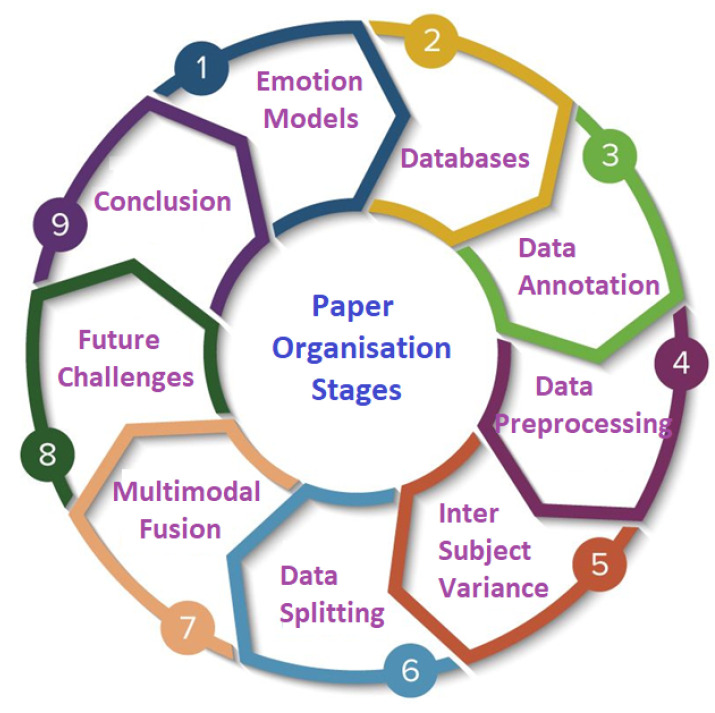
Paper organisation stages.

**Figure 2 bioengineering-09-00688-f002:**
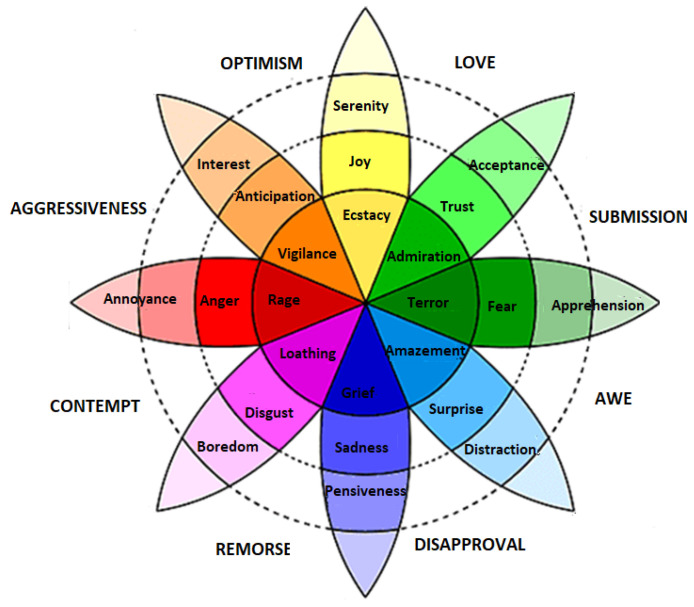
Plutchik wheel of discrete emotions [39].

**Figure 3 bioengineering-09-00688-f003:**
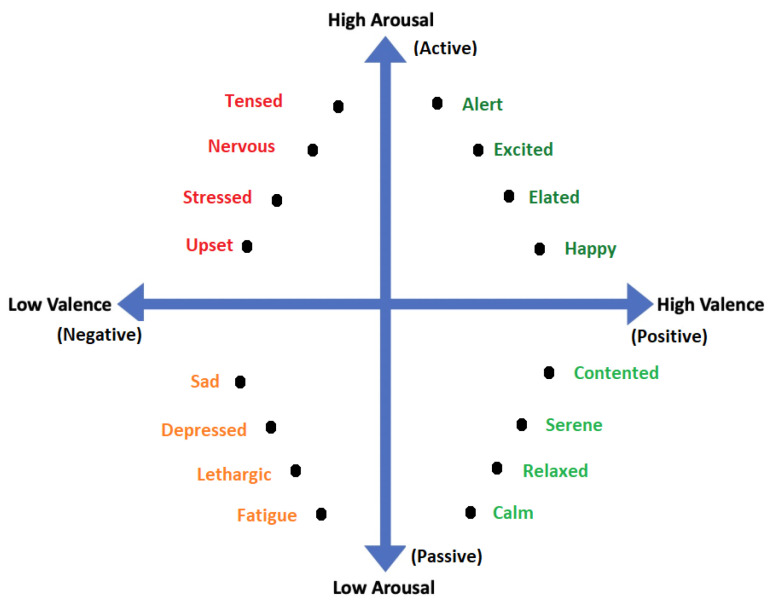
2D Valence-arousal Model.

**Figure 4 bioengineering-09-00688-f004:**
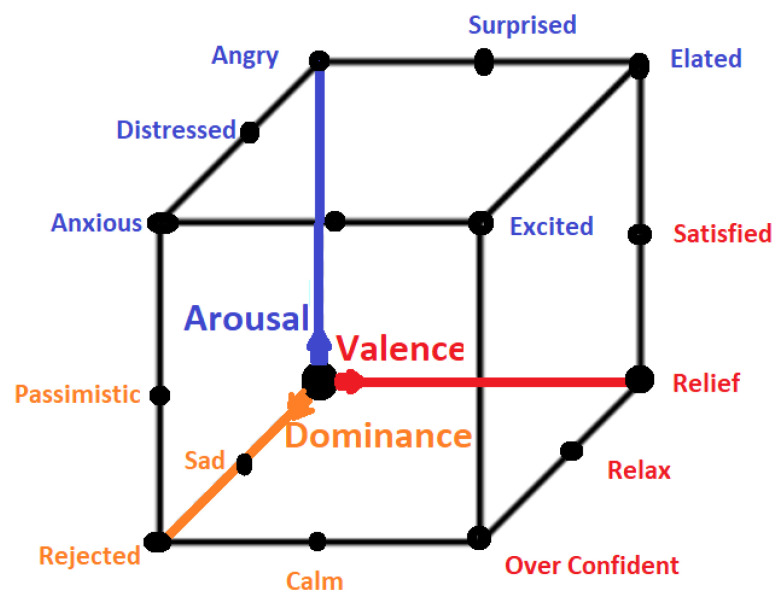
3D Emotion Model.

**Figure 5 bioengineering-09-00688-f005:**
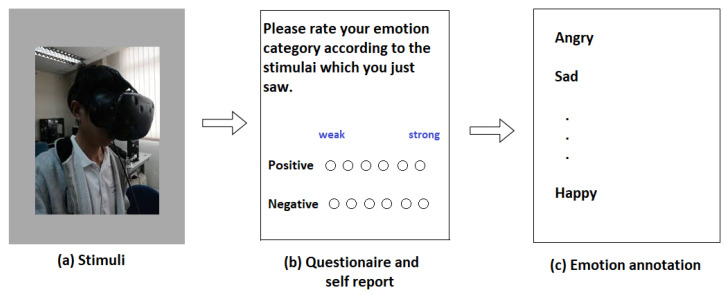
Steps involving discrete annotation.

**Figure 6 bioengineering-09-00688-f006:**
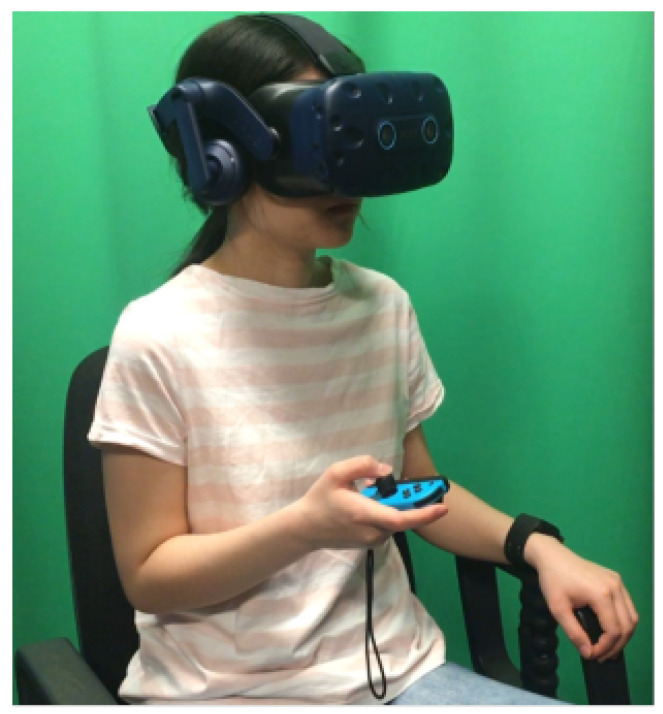
Continuous annotation using HCI mechanism [49].

**Figure 7 bioengineering-09-00688-f007:**
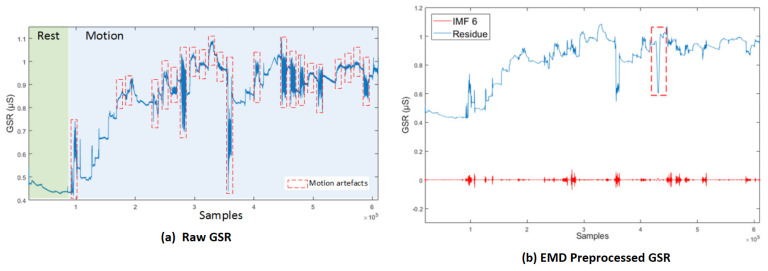
(**a**) Raw GSR: rest and motion phases. Signals corresponding to the movements involving the right hand are delimited by red lines. (**b**) GSR decomposition based on EMD, IMF6 and its respective residue [84].

**Figure 8 bioengineering-09-00688-f008:**
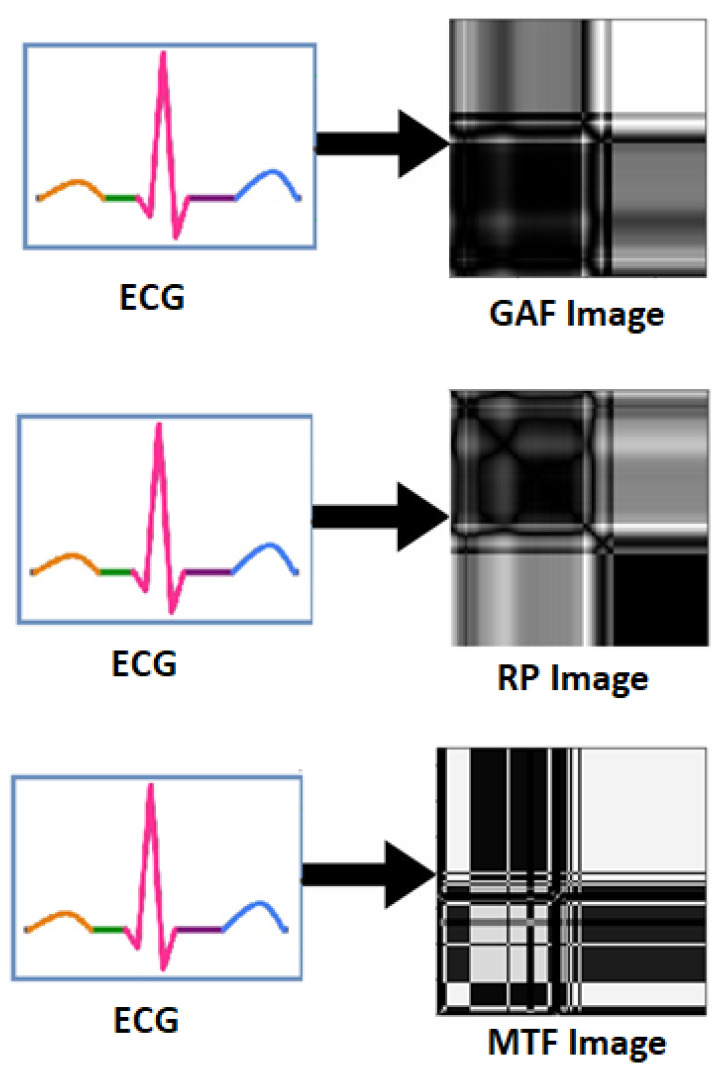
Transformation of ECG signal into GAF, RP and MTF Images.

**Figure 9 bioengineering-09-00688-f009:**
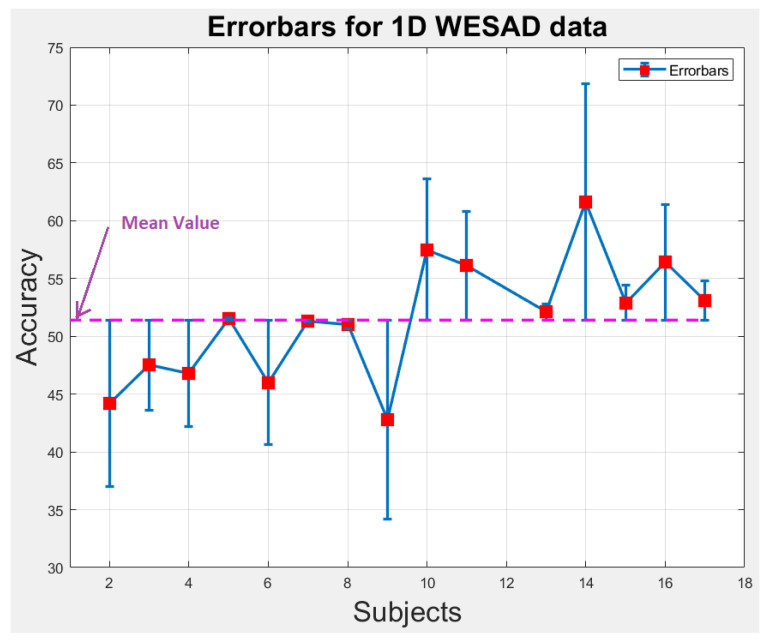
Error-bars showing inter-subject variability in terms of accuracy across the mean value for 1D WESAD data.

**Figure 10 bioengineering-09-00688-f010:**
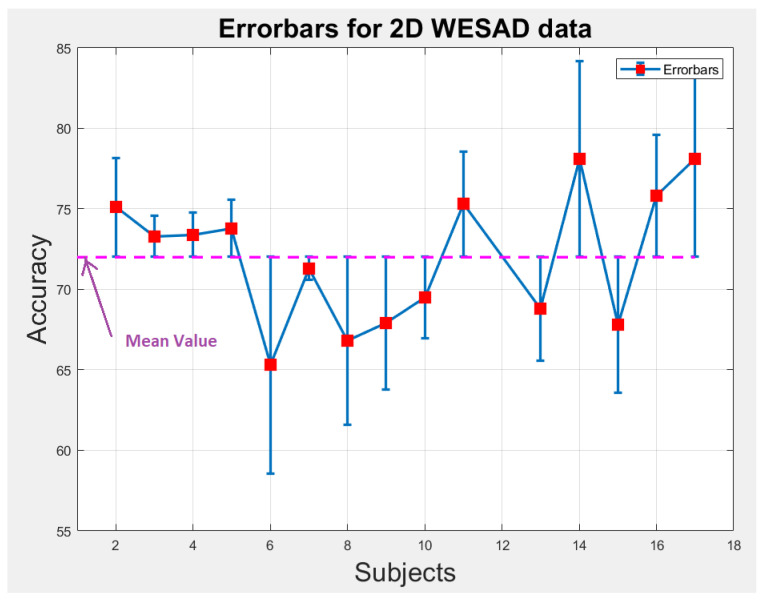
Error-bars showing inter-subject variability in terms of accuracy across the mean value for 2D WESAD data.

**Figure 11 bioengineering-09-00688-f011:**
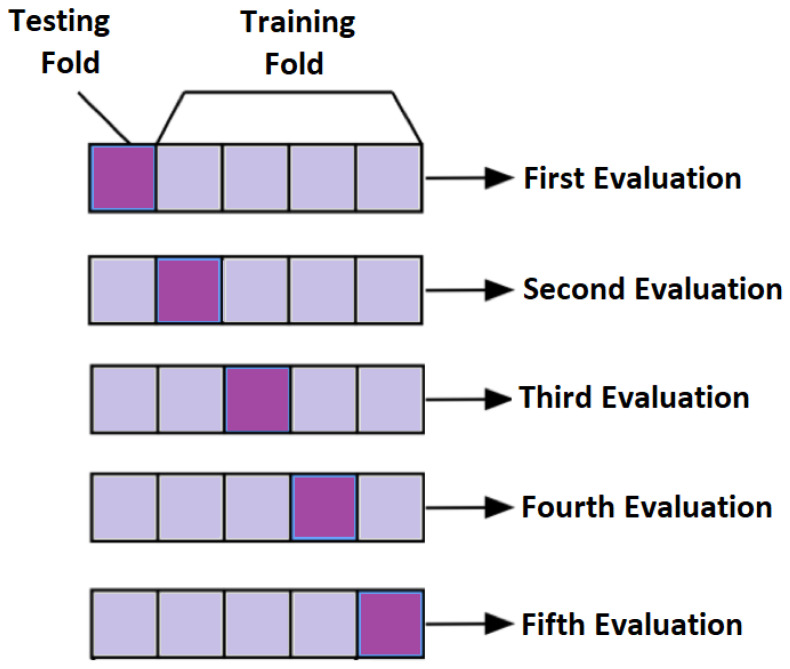
Conceptual Visualization of 5-fold cross validation.

**Figure 12 bioengineering-09-00688-f012:**
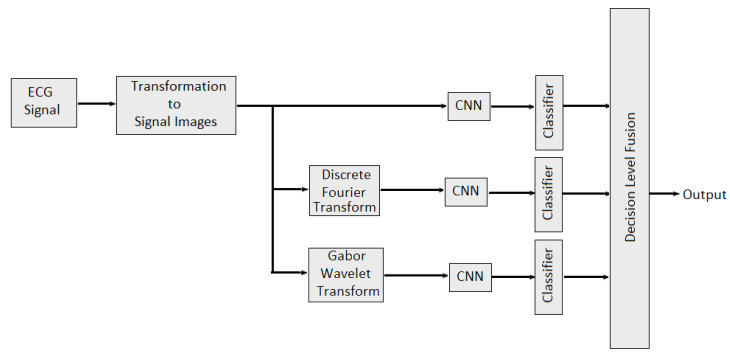
Multimodal Multidomain Fusion of ECG Signal.

**Table 1 bioengineering-09-00688-t001:** Publicly available datasets in alphabetical order for physiological signals-based emotion recognition.

Database	Modalities	Stimulai	Subjects	Sampling Rate	Emotion States
AMIGOS [41]	EEG, ECG, EDA	Video clips	40	Not Provided	Anger, Fear, Sadness, Disgust, Neutrality, Surprise, Happiness
ASCERTAIN [42]	EEG, ECG, EDA	Movie clips	58	EDA at 100 Hz EEG at 32 Hz	Valence (–3–3), Arousal (0–6)
BIO-VID-EMO DB [43]	ECG, EMG, SC	Film clips	86	512 Hz	Valence, Arousal, Amusement, Sadness, Anger, Disgust, Fear
DEAP [44]	EEG, EDA, EMG, PPG, EOG, RSP	Music Videos	32	512 Hz	Valence (1–9), Arousal (1–9), Dominance (1–9), Liking (1–9), Familiarity (1–5)
DREAMER [45]	EEG, ECG	Film clips	23	Not Provided	Anger, Fear, Sadness, Disgust, Calmness, Surprise, Amusement, Happiness, Excitement
MAHNOB-HCI [46]	EEG, ECG, EDA, RSP, SKT	Video clips	27	256 Hz	Anger, Anxiety, Fear, Sadness, Disgust, Neutrality, Surprise, Amusement, Joy
MPED [47]	EEG, ECG, EDA, RSP	Video clips	23	1000 Hz	Anger, Fear, Sadness, Disgust. Neutrality, Funny, Joy
SEED [48]	EEG	Film clips	15	1000 Hz	Negativity, Neutrality, Positivity

**Table 2 bioengineering-09-00688-t002:** Accuracy, precision, recall and $F_{1}$ score using 1D raw WESAD data with 1D CNN.

Testing Sub	Accuracy	Precision	Recall	$F_{1}$ Score
2	44.2	47.3	44.2	40.8
3	47.5	47.9	47.5	45.6
4	46.8	55.4	46.8	43.8
5	51.5	51.3	51.5	49
6	46	45.7	46	43.9
7	51.3	47	51.3	46
8	51	51.4	51	50.6
9	42.8	41.5	42.8	41
10	57.6	56.1	57.6	53.4
11	56.1	57.5	56.1	57.4
13	52.1	51.6	52.1	50.6
14	61.6	60	61.6	55.4
15	52.9	52.1	52.9	50.6
16	56.4	52.5	56.4	52
17	53.1	50.7	53.1	50.9
**Average **	**51.4**	**51.2**	**51.4**	**48.7**

**Table 3 bioengineering-09-00688-t003:** Accuracy, precision, recall and $F_{1}$ score using 2D spectrograms of WESAD data with ResNet-18.

Testing Sub	Accuracy	Precision	Recall	$F_{1}$ Score
2	75.1	75.3	75.1	75.2
3	73.3	75.6	73.3	73.8
4	73.4	76.7	73.4	73.8
5	73.8	77.4	73.8	74.4
6	65.3	67.7	65.3	65.7
7	71.3	76	71.3	72
8	66.8	70.4	66.8	67.7
9	67.9	72.9	67.9	68.4
10	69.5	71.8	69.5	70
11	75.3	78.4	75.3	75.7
13	68.8	73	68.8	69.3
14	78.1	80.1	78.1	78.4
15	67.8	70.6	67.8	68.7
16	75.8	79	75.8	76.1
17	78.1	81.6	78.1	78.2
**Average **	**72**	**75.1**	**72**	**72.5**

**Table 4 bioengineering-09-00688-t004:** Accuracy, precision, recall and $F_{1}$ score using 1D raw RML data with 1D CNN.

Testing Sub	Accuracy	Precision	Recall	$F_{1}$ Score
2	58.9	60.9	58.9	58.6
3	65.4	66.4	65.4	65.4
4	59.3	59.4	59.3	58.9
5	64.7	64.2	64.7	63.2
11	45.3	48.8	45.3	38
12	69.6	72.6	69.6	69.1
13	70.6	76.3	70.6	70.3
14	66.7	69	66.7	67.2
16	64.8	67.2	64.8	64.1
**Average **	**62.8**	**62.5**	**62.8**	**61.6**

**Table 5 bioengineering-09-00688-t005:** Accuracy, precision, recall and $F_{1}$ score using 2D spectrograms of RML data with ResNet-18.

Testing Sub	Accuracy	Precision	Recall	$F_{1}$ Score
2	72.1	74.87	72.1	72.25
3	71.3	75.6	71.25	71.17
4	60.7	58.65	60.7	56.06
5	62.5	63.3	62.5	62.54
11	49.3	37.6	49.3	39.35
12	61.11	63.48	61.11	61.25
13	76.1	83.7	76.15	76.22
14	64	66.13	64	63.59
16	70.3	73.28	70.3	70.8
**Average **	**65.34**	**66.2**	**65.34**	**63.7**

## Data Availability

Not applicable.

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
