# Peer review of "A Survey on Physiological Signal-Based Emotion Recognition"

_bioengineering, 2022, doi:10.3390/bioengineering9110688_

Round 1

Reviewer 1 Report

First of all, I would like to thank you for the opportunity to read your interesting paper entitled “A Survey on Physiological Signal Based Emotion Recognition.”  In this study, authors review the effect of inter-subject data variance on emotion recognition, important data annotation techniques for emotion recognition and their comparison. Furthermore, data preprocessing techniques for each physiological signal, data splitting techniques for improving the generalization of emotion recognition models and different multimodal fusion techniques and their comparison were demonstrated. The paper is very well written. Results were interpreted in light of the proposed research objectives and existing literature. Authors included instructional tables and graphs.

 However, some concerns in your study need to be addressed. I hope my comments below can be helpful for you as you improve this manuscript to deliver its full potential.

Please justify the need for the study in the introduction. The research problem is satisfactorily defined; a little more clarity would do. For that, authors should focus more on addressing what we already know (as we can find, enormous research work has been done on emotion recognition and related issues) about the topic before bringing in a gap considering what the paper tries to fill in. This would make it clear to the reader why it is crucial to address the shortcomings in the literature.

I recommend the authors graphically represent the stages or process of how the paper was organised for better understanding.

The literature review has been satisfactorily done, with room for improvement through minor changes by incorporating more recent, relevant and high-impact articles. Multiple citations should be combined properly.  

The methodology demonstrated is understandable, but needs minor improvement.

Could you please elaborate more in detail on why different methods have been used for EEG preprocessing (i.e. rejection method, linear filtering, statistical methods such as linear regression and Independent Component Analysis) and not the others? What are the drawbacks of others?

The interpretations based on analysis are not quite convincing. More explanation of the results/findings is required.

It would be best if you went a bit deeper into your findings. In many places, you only provided descriptions of results; but you did not give enough explanation of key findings.

The conclusion is a mere synthesis of the research findings. It is underdeveloped, and it falls short in stressing and arguing the original contribution of this research. It should be revised, trying to emphasize the value of this research.

The document is generally of a satisfactory standard. There are minor grammatical or structural errors.

Again, I enjoyed reading your paper and hope my comments can be helpful to you as you improve your manuscript.

Author Response

I have a attached a file

Reviewer 2 Report

The manuscript is focused on the problematic of signals. The manuscript is written in understandable form. It is written on high level and it presented new kinds of information. I have got comments only of technical character. The comments are presented below.

1. Please use similar front in the whole text, also in tables and figures.

2. Please read whole text again, it includes some typographical errors.

3. Authors used too many proceedings papers as literary sources, please replace them by articles from journals.

I hope my comments are helpful.

Author Response

I have attached my response as a word document

Reviewer 3 Report

Dear Authots, thank you very much for this paper.

I describe different limitations.

In particular, emotion recognition is related to age, gender, socio economical level, comorbidity, activity daily living, and marital status.

However, pandemia of Covid-19 is related to higher suicidal %.

Please can you describe the role of these variable on the emotion recognition?

In particular Ciarambino T et al describe that CKD is related to higher depressive symptoms in older age(Nephron Clin Pract

  2007;106(4):c187-92.

Can you report it?

Thank you

Author Response

(The authors gave the same response as above.)

Round 2

Reviewer 1 Report

Dear Authors, I carefully re-evaluated your paper, finding it substantially improved with respect to the version. The revised version is much better organized and has higher scientific quality. Therefore, I recommended it for publication. Thank you

Author Response

We have removed all the potential errors or typos